# EFisioTrack System for Monitoring Therapeutic Exercises in Patients with Shoulder Orthopedic Injuries in a Hospital Setting: A Pilot Feasibility Study

**DOI:** 10.3390/s24154898

**Published:** 2024-07-28

**Authors:** Sergio Hernandez-Sanchez, Jorge Roses-Conde, Neus Martinez-Llorens, Daniel Ruiz, Luis Espejo-Antúnez, Isabel Tomás-Rodríguez, Jose-Vicente Toledo-Marhuenda, Manuel Albornoz-Cabello

**Affiliations:** 1Translational Research Center in Physiotherapy, Department of Pathology and Surgery, Faculty of Medicine, Miguel Hernandez University, Ctra. Alicante-Valencia Km. 8, 7-N 332, 03550 Alicante, Spain; sehesa@umh.es (S.H.-S.); mitomas@umh.es (I.T.-R.); 2Rehabilitation Service, General University Hospital of Elche, Carrer Almazara, 11, 03203 Alicante, Spain; jroses@umh.es (J.R.-C.); neus904@gmail.com (N.M.-L.); 3Department of Computer Technology, University of Alicante, Carrer de Sant Vincent S/N, 03690 Alicante, Spain; druiz@dtic.ua.es; 4Department of Medical-Surgical Therapy, Faculty of Medicine and Health Sciences, University of Extremadura, 06071 Badajoz, Spain; luisea@unex.es; 5Department of Physiotherapy, Faculty of Nursing, Physiotherapy and Podiatry, University of Sevilla, San Fernando, 4, 41009 Sevilla, Spain; malbornoz@us.es

**Keywords:** rehabilitation, shoulder injury, monitoring device, adherence

## Abstract

To assess the effects of the eFisioTrack monitoring system on clinical variables in patients with prescribed physiotherapy for shoulder injuries, twenty-four adult patients with shoulder orthopaedic injuries who underwent physical therapy treatment in a hospital setting participated in the study (twelve in the experimental group and twelve as controls). Clinical outcome measures were shoulder function and pain (Constant–Murley Score and Disabilities of the Arm, Shoulder, and Hand or DASH score). Each variable was measured by a blinded physiotherapist at baseline and at one month follow-up. Patients performed the prescribed exercises either supervised by the physiotherapist (control group) or in a separate room without therapist supervision (experimental group). There were no statistically significant differences between groups before treatment or at follow-up for any outcomes (*p* ≥ 0.05). There was a statistically significant decrease (*p* ≤ 0.05) of at least 10 points in both groups for the DASH score at follow-up. Differences in the total score and subjective components of the Constant–Murley were also evidenced within groups. The use of the eFisioTrack system showed similar results in clinical measures compared to those performed under the direct supervision of the physiotherapist. This approach might be suitable for providing an effective shoulder exercise program at home.

## 1. Introduction

Shoulder pain is a common health problem, and its prevalence has been estimated at around 7–26% [1]. There are several conditions that can produce pain in the shoulder complex, such as fractures, frozen shoulder, rotator cuff tendinopathy, or subacromial syndrome [2]. Physiotherapy management helps to reduce pain, improve range of movement and strength, and improve function. Combining active exercises with manual therapies is a common practice when treating musculoskeletal injuries of the shoulder [3,4].

Therapeutic exercise, in both the physical therapy setting and at home, is a fundamental component of shoulder rehabilitation plans whether treatment is performed with or without surgery, and there is evidence of its effectiveness in the management of shoulder conditions [5]. The primary goal of a shoulder exercise program is to relieve pain, increase strength, reduce muscle imbalances, and restore pain-free joint range of motion [6]. Therefore, its correct execution is of great importance for the clinical course [7].

However, one of the main problems in orthopedic shoulder rehabilitation is poor adherence to home exercise prescriptions [8]. Prescribing a home-based exercise program without the possibility of checking the exercise execution pattern or without a clear element of motivation for compliance can negatively affect clinical results. These factors (lack of supervision, feedback with the therapist, and lack of adherence to treatment) have been strongly considered as possible determinants of poor clinical results for home exercise programs in patients with shoulder pain [9].

Some strategies for their control, such as daily or video recording, have been described, but their use is limited by recall bias or the need for advanced technology [10].

There is a growing number of trials about the use of technology to monitor rehabilitation exercises [11]. Its application in shoulder injuries could be potentially beneficial [12]. Carbonaro et al. [13] presented the development and preliminary testing of a wearable-technology platform for the remote rehabilitation of shoulder muscular-skeletal diseases. This system (Shoulphy) was designed to lead and assess the patient wearing a minimal set of inertial sensors and following personalized physical rehabilitation programs under the remote supervision of the physician/therapist. Pan et al. [14] reported the use of accelerometer-based sensors built into a smartphone to capture the rehabilitation exercises as a self-home monitoring system. Wearable sensors, smartphones, and inertial measurement units have become a feasible option for monitoring joint movement [15,16]. There is also research on the application of Nintendo^®^ Wii technology (Nintendo Wii, Nintendo Co., Ltd., Minami-ku Kyoto, Japan) for monitoring therapeutic exercises on shoulder injuries [17]. This is based on accelerometers and gyroscopes located in the controls, making it a low-cost option. However, most of these focus on stroke, to improve arm function and balance in hemiparetic patients [18,19].

Ruiz-Fernandez et al. [20] described a software platform based on Wii controls technology that could be used to monitor patients’ activity in the scheduled exercise sessions. This platform facilitates the exercise programming with an interesting design for the patient, and also provides feedback between the user and the physiotherapist in real time. These characteristics are differentiating elements from other current rehabilitation methods. However, there is no clinical study of the results obtained in the laboratory. Therefore, the aim of this pilot study was to evaluate the effect on clinical variables of monitoring exercises prescribed for shoulder injury rehabilitation with the eFisioTrack platform in patients of the Rehabilitation Service at University Hospital of Elche.

## 2. Materials and Methods

### 2.1. Study Design

A pilot feasibility study was performed following the recommendations of the CONSORT statement and the Declaration of Helsinki. The study protocol was approved by the Ethics and Research Committee of General University Hospital of Elche (CEIC HGUE-Shs2011) and was prospectively registered in ClinicalTrials.gov (NCT06026137). All participants signed an informed consent form prior to voluntary participation in the study.

### 2.2. Participants

Patients who were referred to the rehabilitation service of the University Hospital of Elche (Spain) for physiotherapy treatment (manual therapy, exercise, stretching, and electrotherapy) after suffering orthopedic injury or surgery in the shoulder joint complex from September 2023 to January 2024 were considered for enrollment in the study. The inclusion criteria were (i) be at least 18 years old and be able to read and understand Spanish; (ii) suffer a traumatic or degenerative shoulder injury, with or without surgical treatment; and (iii) have a prescription for rehabilitative physical therapy that includes active exercises.

Patients were excluded if they had a concomitant injury on an upper extremity or the cervical spine at the time of participation or sequelae of previous injuries in the area. See Figure 1 for a CONSORT diagram of patient selection.

### 2.3. Study Protocol

At baseline, patients were clinically interviewed by a physiotherapist and completed the clinical questionnaires. Patients performed their prescribed physical therapy treatment, at least three times per week in sessions of 45 min in the hospital setting. The treatment involved the application of manual therapy and physical modalities (ultrasound, heat/cold, laser, magnetic field therapy) together with completion of the exercise program for functional recovery of the upper extremity. This program included exercises for muscle strength, scapular stability, joint mobility, and proprioception. In addition, both groups performed the same training load and sets, adapted to the specific condition of each patient, with similar total training time. Figure 2 shows some of the commonly prescribed and selected exercises.

### 2.4. Allocation

Following baseline examination, patients were randomly assigned to either physiotherapist-supervised exercise (control group) or monitoring by the eFisioTrack system (experimental group) using a computer-generated randomization list prepared prior to beginning enrollment (research randomizer: www.randomizer.org).

### 2.5. Intervention

The intervention consisted of the use of the eFisioTrack platform in the experimental group to perform active exercises as part of their shoulder rehabilitation (Figure 2). These were performed independently by each patient in a hospital room, using the eFisioTrack system without supervision by the physiotherapist. The design and use of eFisioTrack has been assessed for usability and was described as appropriate and technically feasible [20]. The subjects were previously instructed in the use of the system in two 20-min sessions. The type of exercise and its parameters were chosen and progressed considering the functional status of the patient and similarity to those executed under the physiotherapist’s supervision.

### 2.6. eFisioTrack Platform

This system is composed of a Wii Remote Wireless Controller, which uses accelerometers to capture trajectories and accelerations. This information is transmitted to a computer via Bluetooth. The processing of the signals received from the remote allows the creation of an intuitive user interface where the patient receives visual feedback about the development of the exercises (number of repetitions, right/wrong, remaining exercises) as well as feedback messages. The software records data on each exercise session, and each user has access to the system, allowing online revision of the patient’s work, both in real time and delayed. Figure 3 represents visually the system components [20,21].

For its use, at the moment of the first patient assessment, with directions from the physiotherapist, a set of “master” exercises was recorded. That is, patients recorded each of them with a specific range of motion and speed (considering dimensional planes and axes of movement), in which the exercises had to be executed correctly.

The physical therapist recorded the prescribed sessions in a calendar: specific exercises and doses (reps and sets) that patients had to perform each day of the week. When patients were performing exercises without therapist supervision, the system compared each repetition of the exercise with their previously recorded “master”.

The system counted each repetition as valid or invalid, comparing them with the recorded trajectory and speed of the “master” exercises (which were continuously revised based on the patient’s evolution). For example, movements performed in a substantially different plane or with a very different speed compared with the recorded master were counted as invalid. A count of repetitions to be performed was shown on a screen, and the system also showed some messages to increase the patient’s motivation during the session, in order to achieve maximal adherence and correct execution (Figure 4).

In addition, the eFisioTrack was able to store the following data for each patient, for each exercise, and for every time: time employed, reps, sets, and a report of mistakes or repetitions not performed correctly.

### 2.7. Outcomes

The following patient-reported outcome measures were used to assess participants’ shoulder pain, function, and health-related quality of life: the Disabilities of Arm, Shoulder and Hand (DASH) score; the Constant–Murley (CM) score; and the 36-Item Short Form Health Survey (SF-36).

The DASH was taken as a primary outcome measure. It is one of the most widely used self-reported questionnaires that measures symptoms and degree of function related to a disorder in the upper extremity [22]. It has been validated in Spanish [23] and comprises 30 items: 21 about physical function, 6 for symptoms, and 3 to assess social aspects. In addition, there are two optional modules, each with four items, which are used to assess symptoms and function in those whose functional demands are not included in the main part of the questionnaire.

This instrument has been used in previous studies involving physical therapy and exercise for shoulder injuries [24]. It is scored in two components: first, the symptom questions (30 items), and second, the optional modules. The assigned values for all completed responses are summed and averaged, producing a score that is then transformed on a scale of 0 to 100 by subtracting 1 and multiplying by 25. A higher score indicates greater disability. Differences in scores are considered clinically relevant (minimum clinically important difference or MCID) when they are above 10 points [25].

The CM score is a commonly used specific instrument for assessing the shoulder joint [26]. The maximum score is 100 points, with 90 to 100 being excellent, 80 to 89 good, 70 to 79 medium, and less than 70 poor, considering that the scores can vary with age. This tool includes a subjective assessment of the patient’s pain and ability to perform daily activities (35 points) and an objective assessment of mobility and strength by physical examination (65 points). Its use has been specifically validated in shoulder arthroplasty, rotator cuff repair, adhesive capsulitis, and fractures of the proximal humerus, but not in shoulder instability [27].

The SF-36 is a generic measure of health-related quality of life, with 36 questions. It yields an 8-scale profile of functional health and well-being scores that can be divided into two categories, mental and physical. The value range is between 0 and 100 (higher scores indicating better health status) and has been validated in Spanish [28].

### 2.8. Data Collection

All participants were assessed at baseline on their first visit to the physical therapy area and at one-month follow-up. The assessments were conducted by the same physical therapist (N.M.), who did not participate in the treatments and was blinded to the study groups.

### 2.9. Statistics

Descriptive data are presented as a mean and standard deviation for continuous variables. For statistical analyses, the normality of the data distribution was assessed by the Shapiro–Wilk test. To compare differences between groups in the studied variables, Student’s *t*-test was used for normally distributed data. For each group, a paired *t*-test was used to compare baseline and follow-up scores, and the 95% confidence interval (CI) was calculated for the mean differences. Effect size (ES) was estimated using Hedges g with the following scale to categorize the magnitude of this ES: <0.2 = trivial; 0.2–0.5 = small; 0.5–0.8 = medium; 0.8–1.3 large; and >1.3 very large [29].

The theoretical sample size based on a 10-point difference (10%) in the DASH between the two studied groups, assuming a 95% confidence interval, 80% statistical power, and 20% sampling error, resulted in 36 patients in each group. This pilot study included a smaller sample for exploratory purposes.

All analyses were conducted using SPSS Statistical Package for Windows, version 25 (SPSS Inc., 2009, Chicago, IL, USA). The significance level was set at *p* < 0.05.

## 3. Results

Thirty consecutive patients with shoulder injuries and with an exercise prescription in their physical therapy treatment were screened for the eligibility criteria. Twenty-four subjects (mean ± SD age, 57.1 ± 6.3 years) met the inclusion criteria and were randomly assigned to the experimental (*n* = 12) or control (*n* = 12) groups. At one-month follow-up, analyses were conducted on eleven subjects of the control group and twelve subjects of the experimental group. Baseline characteristics of this sample are shown in Table 1. The patients’ diagnoses appear in Table 2. 

There were no statistically significant differences between the groups for any of the registered outcome measures (Table 3). With respect to the DASH score, there was a statistically significant reduction of 11.3 and 10.7 points for the experimental and control groups, respectively, at follow-up. For the CM score, there were also statistically significant changes within-group between baseline and one-month follow-up assessments.

Considering individual parameters of the CM, significant differences were registered in the subjective data sections: pain and activities of daily living for both groups with large effect sizes (Table 4). With respect to the shoulder range of movement and muscle strength measured by this scale (parameters 3 and 4), no significant differences were found either between groups nor within group.

## 4. Discussion

The aim of this pilot study was to compare clinical outcomes at one-month follow-up in subjects with shoulder injuries who received physiotherapy treatment and used a remote monitoring system (eFisioTrack) for their exercise program versus patients who performed the exercise program supervised by the physical physiotherapist in a hospital room.

Regarding clinical outcome measures, there was an improvement in functionality, measured on the DASH scale, and no differences between groups were found at follow-up. On the basis of these results, it seems that a four-week non-supervised exercise program, adjunct to conventional physiotherapy treatment, did not show statistically significant differences with respect to the inpatient scenario in a supervised exercise group. This suggests that patients could perform their autonomous exercise sessions without direct face-to-face supervision from the therapist, with better resource efficiency.

The DASH scale, used to measure the functional capacity of the shoulder, has been demonstrated to be a reliable and valid measure of disability in those complaining of upper extremity dysfunction [30] and has been used in previous studies involving manual therapy and exercise for shoulder impingement syndrome [24]. For this study, only the general part of the scale has been used, not the specific modules, because most of the subjects were on sick leave or unemployed. The minimum clinically important difference (MCID) value according to the DASH has been reported as between 10.8 and 15 points [25]. In this study, both the control group and the experimental group achieved a reduction of at least 10 points on the DASH score (10.7 and 11.3 points, respectively), indicating a slight but significant clinical improvement, although not statistically significant. A higher number of patients and a longer follow-up period is required to assess this variation consistently and to check if it is maintained in the long term.

With respect to the changes in the CM score, both groups showed an improvement: the CM score was higher in the experimental group than in the control group (7.75 and 4.9 points, respectively), and the difference between group change scores was 3.2 points at follow-up. Moreover, in the analysis of the CM score per component, there was a statistically significant improvement in the subjective variables pain and activities of daily living (ADL). The results in the experimental group were equal to or better than those in the control group. The movement component also showed an improvement in the experimental group compared to the control group, although it was not significant. Regarding the strength component, no improvement was observed in the experimental group, and there was even a slight loss in the control group (0.1 and −0.4 points, respectively). All these changes are similar to those observed in another preliminary study carried out by Pastora-Bernal et al. [31] to evaluate the feasibility and effectiveness of a telerehabilitation program versus traditional care along 12 weeks. In this case, the CM score showed significant improvements in both groups, a difference up to 20 points after 12 weeks, but no difference was found between the two rehabilitation methods. Dias Correia et al. [32] studied the effect of a hybrid protocol, based on an assisted technology therapy program combined with face-to-face sessions, versus a face-to-face-only program. Although no differences were found between groups in the QuickDASH nor the CM, the results showed better scores in the digital therapy group for QuickDASH, as well as an interaction between time and group in the CM score. All these results suggest that a telerehabilitation program with range of motion, strengthening, and scapula stabilization exercises, after an arthroscopic rotator cuff repair, seems to be similar rather than inferior to traditional face-to-face physiotherapy.

Although the CM score is one of the most widely used tests for functional outcome shoulder assessment, it has several weaknesses. For example, the assessment of strength is one of the non-standardized aspects of the instrument and the one that generates the most discrepancies [33]. A definition of the exact maneuver to measure force has never been given, nor has a specification of the exact position of the arm or location of resistance been given [26].

For the assessment of strength in the CM score in the present research, the method of incremental free weights was used with the subject standing and the shoulder in the position of maximum abduction [34]. The weight that the patient could lift and hold for 5 s was considered, but it was not sensitive enough to determine the level of force. Given its relative weight in the global score (25/100), the low sensitivity of the method could have influenced the total scores, this being a possible limitation of the results.

Supervised exercise prescription is a well-established and effective measure for the treatment of several shoulder disorders [35,36]. This active strategy requires time and close attention from the therapist [26]. In recent years, and especially since the COVID pandemic, there has been an increase in studies that assess the effects of telerehabilitation compared with face-to-face physiotherapy under different conditions [37,38]. There are sensors based on accelerometers and/or gyroscopes that are wildly used for movement assessment and recognition [39,40]. Considering the need for a low-cost system and the patient’s autonomy and comfort, we chose a gamepad manufactured by Nintendo. The Wii Remote© has been used for general purpose gesture recognition and in few rehabilitation projects as well, but they are focused on testing different modes of virtual rehabilitation, especially in carrying out prescribed exercises in strokes or other conditions, without paying too much attention to the communication and monitoring of the process [18,19,41,42]. Instead, this software platform allows feedback between the user and the physiotherapist in real time, which is considered an element that differentiates it from other telerehabilitation systems.

In spite of the reduced sample size, the preliminary results obtained allow some conclusions to be drawn. Regarding adherence to treatment, eFisioTrack can improve it in different ways: (i) Customized movement, which is considered as the master movement for a specific patient and exercise, is recorded; (ii) patients and physiotherapists have access at any time to all the data related to the rehabilitation plan and can even obtain real-time feedback during the movement; (iii) it allows contact with their physiotherapist and an easy adjustment of the patient’s progress; (iv) the recording of the data allows the physiotherapist to know if the work routine has been completed or not and to what extent, which maintains patient adherence.

The relationship between adherence and the place where the rehabilitation treatment is carried out, the outpatient or inpatient scenario, has been widely studied. Several studies recorded high non-adherence to physical rehabilitation therapies in outpatient settings, reaching around 70% [43,44]. The intention to engage in the home exercise program, self-motivation, self-efficacy, previous adherence to exercise-related behaviors, social support from friends and family, and effective communication with care providers have been identified as factors predicting adherence [43].

The attractiveness of exercise programs, the feedback model used, and recent technologies make the routine more interesting for patients but the possibility of being guided while performing exercises must be present in any case, regardless of the model used [45]. The score of a qualitative study conducted by Ruiz-Fernámdez et al. [20] on the perception and satisfaction of users regarding management of the eFisioTrack was high, reaching values between 4.4 and 5 out of 5 points.

The system shows promising results as a work item in shoulder home-based rehabilitation. In fact, real-time telerehabilitation is a strategy for interventions in other musculoskeletal conditions, to provide continuity to healthcare services and mitigate distance and displacements [46]. Azure Kinect has also been used as a telerehabilitation platform in the shoulder motor function recovery, but, in contrast with eFisioTrack, is based on a set of serious games to increase the patients’ engagement, and it does not have many of the functions that are present in eFisioTrack [47].

The systematic review carried out by Gava et al. [48] about the effects of physical therapy given by telerehabilitation in patients with shoulder pain and disability found a small number of randomized controlled trials investigating their use. Six studies were included with similar injuries to those in our intervention (shoulder post-operative care, chronic shoulder pain, and frozen shoulder), but although telerehabilitation may be a promising tool, the studies presented a very low quality of evidence and a definite recommendation of the use of telerehabilitation in this population is not well supported. An important aspect to consider is that only one study investigated the effect of synchronous telerehabilitation through videoconferencing compared to a home exercise program [49]. In this case, a statistically significant difference in favor of telerehabilitation was obtained in relation to pain and disability in patients undergoing shoulder joint replacement.

eFisioTrack allows the therapist to schedule a hybrid intervention combining in-person interventions and synchronous or asynchronous telerehabilitation interventions, according to the needs of the patient, in each case. Furthermore, the system can establish measures to control and increase adherence, something essential to achieving good results [50], since the Wii Remote pad can register alterations in the execution of the exercises (trajectory, speed, repetitions) and help make decisions in this regard.

Despite the lack of studies with the Wii Remote pad for rehabilitation of shoulder injuries, and that most of them have been executed on painful hemiplegic shoulder in stroke patients [18,19], we believe that it is an adequate, inexpensive, and easy-to-use device that, associated with the appropriate platform, can offer a good telerehabilitation service. Cost analysis for the eFisioTrack should be considered in the future, as has been done in other pathologies such as low back pain, hip and knee osteoarthritis, and total knee arthroplasty [41,42,43,44,45,46,47,48,49,50,51,52,53,54,55]. In these studies, the use of telerehabilitation has shown positive clinical outcomes, with a fewer admission to hospitals, shorter length of stay in hospitals, lower number of visits to the rehabilitation service and, consequently, lower costs.

Given the high prevalence of shoulder injuries, telerehabilitation based on the eFisio Track system represents an opportunity to improve clinical and economic data. In addition, the extensive use of the system as a complement to a home therapeutic exercise program could have a positive impact on adherence to performing the exercises. This aspect could potentially improve clinical and functional results.

If this were achieved, its implementation could contribute to reducing costs in the rehabilitation of orthopedic shoulder injuries through fewer patient trips to the hospital environment, and a lower consumption of human resources dedicated to supervising the execution of the exercises.

In our pilot study, carried out in an inpatient scenario but designed to be performed at the patient’s home, the execution of exercises without direct supervision of the physiotherapist does not seem to be different from in-person physical therapy to improve shoulder pain and disability outcomes, which is consistent with other similar studies [31,32].

Confirmation of these preliminary results could reduce the workload of physiotherapy services, contributing to resource efficiency [56]. Nevertheless, the results of this study must be interpreted with caution due to some limitations. In the first place, the small sample size is justified because some patients were dissatisfied with a rehabilitation program in an empty hospital ward, without a physiotherapist, and preferred a more direct treatment model. The analyzed sample includes an age range of 45–67 years, since our goal for the user study was to inform design and assess feasibility. Despite the fact that it is a preliminary study, we cannot fail to point out that there will be an effect in a broader age range with a larger sample size. The follow-up period of one month has probably been too short to evaluate stable results for this type of injury that usually have a mid- or long-term evolution [57].

Another important aspect that must be considered is that the effect cannot be only attributed to the application of the system because there are variables that are not controlled, such as the influence of the therapist or the interface, and the natural course of the pathology, among others. Finally, the fact that the patient had to travel to the hospital, instead of performing the exercises at home, with variables associated with a clinical setting, may have influenced the results. The same methodology must be extrapolated to the home environment, which is our intention in the future.

## 5. Conclusions

The results of the current study suggest that an exercise program using the eFisioTrack system, combined with manual therapy and physical modalities, is associated with clinical improvements in shoulder pain and disability at one-month follow-up. The results are at least equal to those obtained with the same protocol under supervision by the physiotherapist during exercise execution. The use of the eFisioTrack monitoring system could facilitate the execution of the prescribed shoulder therapeutic exercise programs without detriment to the clinical outcomes.

## Figures and Tables

**Figure 1 sensors-24-04898-f001:**
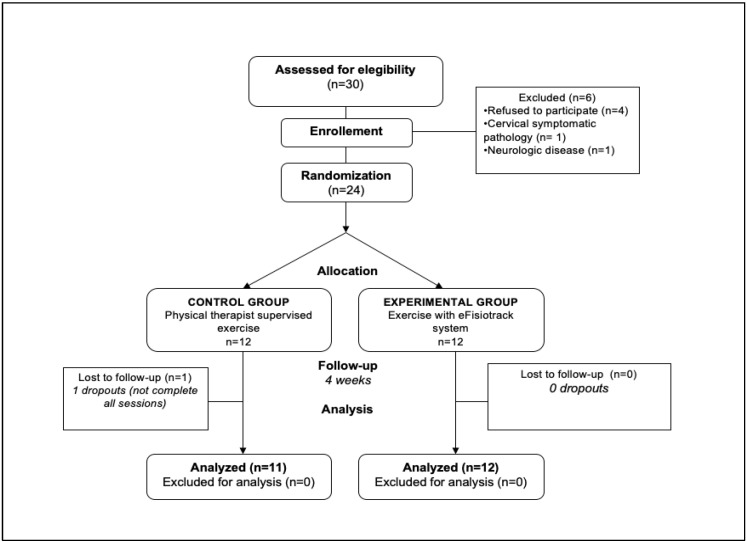
CONSORT flow diagram of patient selection.

**Figure 2 sensors-24-04898-f002:**
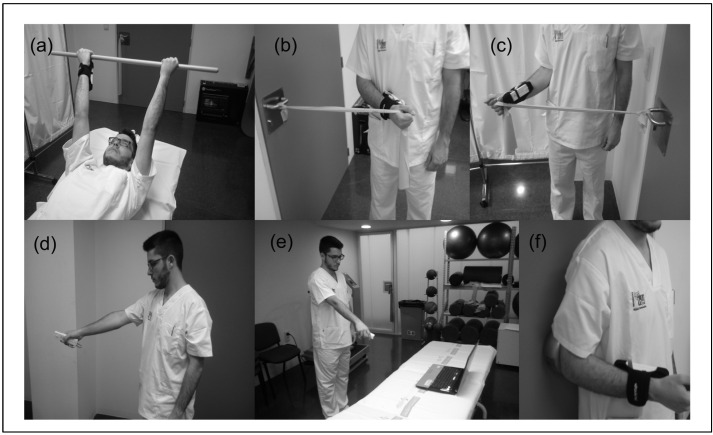
Examples of the commonly prescribed and selected exercises. (**a**) Active bilateral shoulder flexion; (**b**,**c**) internal and external rotation resisted exercises with elastic bands; (**d**) active shoulder abduction; (**e**) general view of the eFisioTrack scenario; (**f**) isometric shoulder extension.

**Figure 3 sensors-24-04898-f003:**
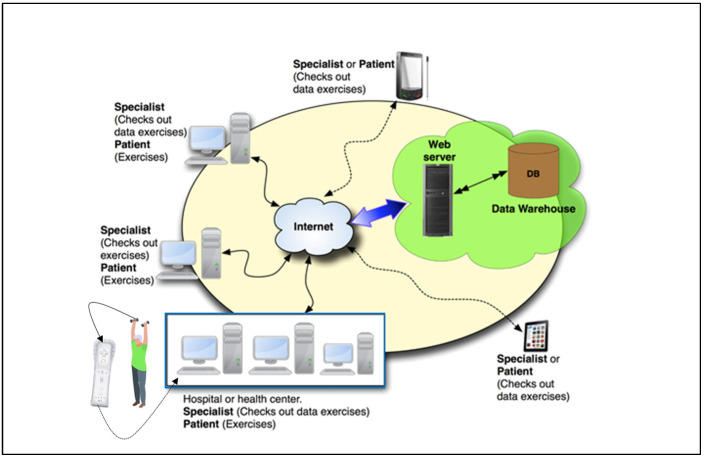
eFisioTrack system components.

**Figure 4 sensors-24-04898-f004:**
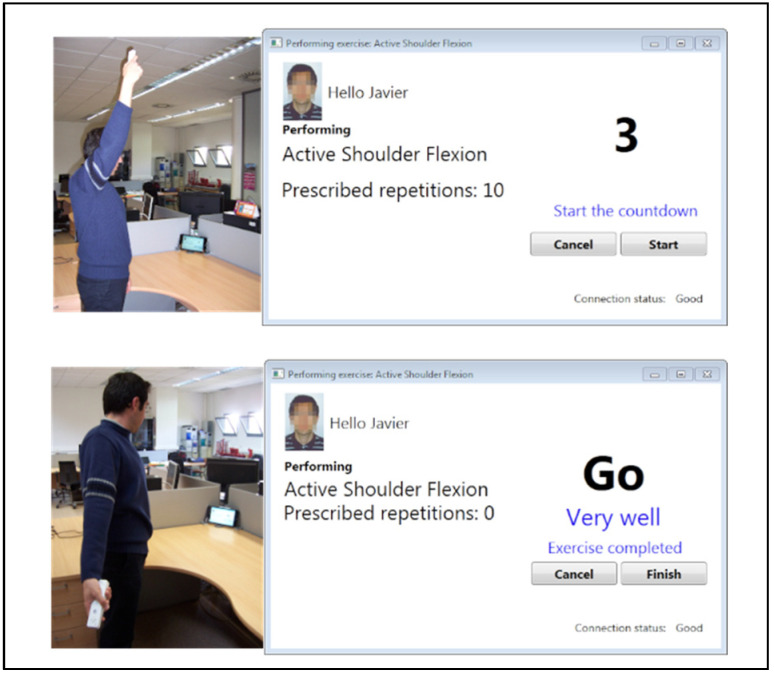
Screenshot of the eFisioTrack user interface.

**Table 1 sensors-24-04898-t001:** Baseline demographics for both groups.

	Experimental (*n* = 12)	Control (*n* = 11)	*p* Values
Age, years	56.5 ± 1.7	57.8 ± 5.6	0.222
Gender, male/female	6/6	4/7	
Height, m	1.6 ± 0.1	1.6 ± 0.1	0.264
Weight, kg	62.5 ± 4.8	66.8 ± 7.3	0.093
Body Mass Index (kg/m^2^)	24.0 ± 1.1	24.8 ± 1.9	0.070
Constant score	38.4 ± 4.9	38.0 ± 6.4	0.446
DASH	44.2 ± 3.7	41.2 ± 5.8	0.080
SF-36 Physical	40.7 ± 15.4	38.5 ± 5.8	0.458
SF-36 Mental	51.7 ± 6.6	49.7 ± 10.5	0.389
Dominant side			0.764
Right	8	7	
Left	4	4	
Affected side			0.764
Right	7	6	
Left	5	5	

BMI, body mass index; DASH, Disability of the Arm, Shoulder, and Elbow questionnaire: SF-36, 36-Item Short Form Health Survey.

**Table 2 sensors-24-04898-t002:** Diagnosis of the included patients.

Diagnosis	Number of Patients
Experimental group (*n* = 12)	
Proximal humeral fracture	3
Adhesive capsulitis	2
Head humerus fracture	1
Subacromial syndrome	6
Control group (*n* = 11)	
Proximal humeral fracture	2
Adhesive capsulitis	1
Head humerus fracture	1
Subacromial syndrome	7

**Table 3 sensors-24-04898-t003:** Mean differences between groups and within-group for all clinical variables.

	Baseline	Follow-Up	Within-Group Change Scores	Between-Groups Change Scores
**Constant total score**				3.2 (−1.1; 7.5)
Control	38.0 ± 6.4	42.9 ± 5.5	4.9 (2.8; 7.0) * (ES = 3.364)	
Experimental	38.4 ± 4.9	46.21 ± 4.4	7.7 (6.4; 9.1) * (ES = 2.309)	
**DASH score**				2.5 (−1.8; 6.8)
Control	41.2 ± 5.8	30.5 ± 5.1	−10.7 (−11.6; −9.8) * (ES = 1.394)	
Experimental	44.2 ± 3.7	32.9 ± 4.8	−11.3 (−12.5; −10.1) * (ES = 2.053)	
**SF-36 Physical**				2.2 (−1.3; 5.9)
Control	38.5 ± 5.8	36.5 ± 3.5	−2.0 (−3.9; −1.2)	
Experimental	40.7 ± 15.4	38.7 ± 12.7	−2.0 (−6.8; −0.3)	
**SF-36 Mental**				4.3 (−1.2; 7.5)
Control	49.7 ± 10.5	46.4 ± 11.8	−3.3 (−4.5; −0.2)	
Experimental	51.7 ± 6.6	50.7 ± 10.1	−1.0 (−3.1; −0.3)	

DASH, Disability of the Arm, Shoulder, and Elbow questionnaire: SF-36, 36-Item Short Form Health Survey. * *p* < 0.01.

**Table 4 sensors-24-04898-t004:** Mean differences between groups and within-group for each parameter of the Constant–Murley scale.

	Baseline	Follow-Up	Within-Group Change Scores	Between-Group Change Scores
**Pain**				0.4 (−0.57; 1.4)
*Control*	4.7 ± 1.3	6.2 ± 1.2	1.5 (1.2; 1.9) * ES = 0.615	
*Experimental*	4.6 ± 0.8	6.6 ± 1.0	2.0 (1.5; 2.5) * ES = 0.826	
**ADL**				0.7 (−0.4; 1.7)
*Control*	8.9 ± 1.8	10.8 ± 1.5	1.9 (1.1; 2.7) * ES = 1.231	
*Experimental*	8.7 ± 1.1	11.5 ± 1.0	2.8 (2.4; 3.3) * ES = 0.771	
**Movement**				1.5 (−1.6; 4.5)
*Control*	20.6 ± 3.9	22.5 ± 4.0	1.5 (0.5; 3.3)	
*Experimental*	21.2 ± 3.3	24.0 ± 3.0	2.8 (1.9; 3.8)	
**Strength**				
*Control*	3.8 ± 0.7	3.4 ± 0.8	−0.4 (−0.8; −0.1)	0.6 (−0.2; 1.6)
*Experimental*	3.9 ± 1.0	4.0 ± 1.1	0.1 (−0.1; 0.3)	

Values are expressed as mean ± SD for baseline and follow-up and as mean (95% confidence interval) for within- and between-group change scores. ADL: activities of daily living. * *p* < 0.01. ES, effect size.

## Data Availability

The original contributions presented in the study are included in the article; further inquiries can be directed to the corresponding author.

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
