# Peer review of "EFisioTrack System for Monitoring Therapeutic Exercises in Patients with Shoulder Orthopedic Injuries in a Hospital Setting: A Pilot Feasibility Study"

_sensors, 2024, doi:10.3390/s24154898_

Round 1

Reviewer 1 Report

Comments and Suggestions for Authors

This study assessed the effect of the eFisioTrack monitoring system on clinical variables in patients with prescribed physiotherapy for shoulder injuries. Using statistical methods, the authors analyzed the clinical data of 24 patients and concluded that the results obtained using eFisioTrack monitoring system were almost identical to those obtained using the same regimen under the supervision of a physical therapist. However, the sample size of this study was small and the study period was short. I suggest that the sample size and research period can be further expanded, so as to expand the depth of this study and improve its credibility. I think this work could be published in Sensors journal. Before that, I hope the authors will reply to the following questions and make some modifications.

1. Is there a significant difference in the total training time of patients using eFisioTrack system compared with patients in the control group? If so, then this may also be an important factor in the rehabilitation effect. It is suggested that the authors provide this part of data and make a discusssion.

2. Some information is not clear. For example, why is the total number of people in the "Dominant side" in Table 1 not equal to 12? In addition, what are "SF-36physical" and "SF-36mental" in Table 3.

3. “A flow chart diagram of patient selection, along with reasons for ineligibility and dropouts are shown in figure 2.” This should be figure 1.

4. The subjects of experimental group in this study were also treated in the hospital environment, but the content of the study was the effect of patients using eFisioTrack system at home. In subsequent study, subjects should be treated at home to make the results more realistic.

Comments on the Quality of English Language

None

Author Response

 Comments 1

This study assessed the effect of the eFisioTrack monitoring system on clinical variables in patients with prescribed physiotherapy for shoulder injuries. Using statistical methods, the authors analyzed the clinical data of 24 patients and concluded that the results obtained using eFisioTrack monitoring system were almost identical to those obtained using the same regimen under the supervision of a physical therapist. However, the sample size of this study was small and the study period was short. I suggest that the sample size and research period can be further expanded, so as to expand the depth of this study and improve its credibility. I think this work could be published in Sensors journal. Before that, I hope the authors will reply to the following questions and make some modifications.

  1. Is there a significant difference in the total training time of patients using eFisioTrack system compared with patients in the control group? If so, then this may also be an important factor in the rehabilitation effect. It is suggested that the authors provide this part of data and make a discussion.

Thanks for the comment. It is true that the manuscript does not provide information related to that topic and, as you point out, total training time of patients, depending on the group, may also be an important factor in the rehabilitation effect. However, we understand that it is not appropriate to expand the discussion section with this topic because in our study, both groups performed the same training load with similar exercise times, without significant differences.

The “material and methods” section does not adequately explain this consideration and for this reason we have modified the text, in the study protocol section, as follows:

Line 147 to 149

“In addition, both groups performed the same training load and sets, adapted to the specific condition of each patient, with similar total training time”.

  1. Some information is not clear. For example, why is the total number of people in the "Dominant side" in Table 1 not equal to 12? In addition, what are "SF-36physical" and "SF-36mental" in Table 3.

2.1. After reviewing the comment and the table 1, we confirm that there was an error with these data. In accordance with the reviewer's comment, we have modified the information of table 1.

2.2. Regarding the comment and the table 3.

The Short Form-36 Health Survey (SF-36) is one of the most widely used and evaluated generic health-related quality of life (HRQL) questionnaires. It was developed in the USA and has been translated into 120 languages and is commonly used to provide information of the health status of different populations, in order to measure the impact on health of clinical and social interventions.

It is a 36-item scale, which measures eight domains of health status: physical functioning (10 items); physical role limitations (four items); bodily pain (two items); general health perceptions (five items); energy/vitality (four items); social functioning (two items); emotional role limitations (three items).

The SF-36 physical and SF-36 mental are the standardized components of the categories in which the 8 domains represented in the SF36 questionnaire are grouped.

After reviewing the comment of the reviewer and the manuscript, we have added information of the Short Form-36 Health Survey (SF-36) as shown below. In addition, we have included a new reference.

Line 264 to 266

“The following patient-reported outcome measures were used to assess participants’ shoulder pain, function and health-related quality of life: the Disabilities of Arm, Shoulder and Hand (DASH) score; the Constant-Murley (CM) score; and the Short Form-36 Health Survey (SF-36)”.

Line 290 to 293

“The SF-36 is a generic measure of health-related quality of life, with 36 questions. It yields an 8-scale profile of functional health and well-being scores that can be divided in two categories - mental and physical. The value range is between 0 and 100 (higher scores indicating better health status) and has been validated in Spanish [28]

[28] Alonso, J.; Prieto, L.; Anto, J.M. The Spanish version of the SF-36 Health Survey (the SF-36 health questionnaire): an instrument for measuring clinical results.  Med Clin (Barc).1995,104,771– 776.  

  1. “A flow chart diagram of patient selection, along with reasons for ineligibility and dropouts are shown in figure 2.” This should be figure 1.

Thanks. Effectively, the number of the CONSORT flowchart diagram is figure 1. This has been changed in the manuscript.

  1. The subjects of experimental group in this study were also treated in the hospital environment, but the content of the study was the effect of patients using eFisioTrack system at home. In subsequent study, subjects should be treated at home to make the results more realistic.

We absolutely agree with this suggestion and that issue pointed will be the next level of the study.

Once we have check that the use of the Efisiotrack has shown non-inferiority results in comparison with supervised training, the next step is to develop a new study using the system at home. In fact, this proposal for the future is indicated in the final section of the manuscript, just before the conclusions section (lines 605-606):

"The same methodology will be extrapolated to the home environment, which is our intention in the future”.

However, there are certain technical and equipment requirements to consider (availability of a computer by patients at home; transfer them by the research team; internet access, or transfer of a wireless router; and a training of the patients in the use of the system during in-person sessions). All of these needs require additional funding and we are working on it.

Extra clarification

After submitting the manuscript to your journal, we received an email from Ms. Elsie Zhao - Managing Editor - advising us to review the current Data Availability Statement in our manuscript and make the necessary adjustments to guarantee the quality and transparency of the research published in Sensors.

In this sense, we have added a file in supplementary material with the database of our study, and we have modified this statement in the manuscript, as shown below.

Line 631 to 632

Data Availability Statement: The original contributions presented in the study are included in the article/supplementary material, further inquiries can be directed to the corresponding author/s.

Reviewer 2 Report

Comments and Suggestions for Authors

Efisiotrack system for monitoring therapeutic exercises in patients with shoulder orthopedic injuries in a hospital setting: a pilot feasibility study

Comments:

This study evaluated the eFisioTrack system for monitoring therapeutic exercises in patients with shoulder injuries. Twenty-four patients were divided into two groups: one using eFisioTrack and the other with physiotherapist supervision. Both groups showed significant improvements in shoulder function and pain, measured by the DASH and Constant Murley scores, with no significant differences between the groups. The eFisioTrack system demonstrated similar effectiveness to supervised therapy, suggesting it could be a viable option for home-based rehabilitation, improving resource efficiency and patient adherence. Further studies are recommended to confirm these findings and explore broader applications.

1. The introduction is comprehensive and provides a good background on the prevalence and impact of shoulder pain. However, it could benefit from a more detailed discussion on the limitations of current rehabilitation methods to better justify the need for the eFisioTrack system.

2. The statistical analysis section is thorough. Consider adding more details on the clinical significance of the findings, not just the statistical significance.

3. The description of the eFisioTrack system is comprehensive. Consider adding a figure or diagram to represent the system and its components visually.

4. The chosen outcome measures are appropriate. It would be beneficial to provide a brief explanation of why these specific measures were chosen.

5. Discuss the broader implications of the study for clinical practice and future research. Consider including a paragraph on the potential economic benefits of the eFisioTrack system.

Comments on the Quality of English Language

The quality of the English language in the manuscript is quite good

Author Response

Comments 2

This study evaluated the eFisioTrack system for monitoring therapeutic exercises in patients with shoulder injuries. Twenty-four patients were divided into two groups: one using eFisioTrack and the other with physiotherapist supervision. Both groups showed significant improvements in shoulder function and pain, measured by the DASH and Constant Murley scores, with no significant differences between the groups. The eFisioTrack system demonstrated similar effectiveness to supervised therapy, suggesting it could be a viable option for home-based rehabilitation, improving resource efficiency and patient adherence. Further studies are recommended to confirm these findings and explore broader applications.

Thanks for this comment. Of course, these findings are only preliminary and future studies are required. The project is designed to try to improve the resource efficiency and patient adherence as pointed, but this requires studies with other designs and methodology that we are planning to develop.

  1. The introduction is comprehensive and provides a good background on the prevalence and impact of shoulder pain. However, it could benefit from a more detailed discussion on the limitations of current rehabilitation methods to better justify the need for the eFisioTrack system.

We discussed that topic in the introduction (lines 52 to 92), but we have added text, as well as a new reference, to more appropriately underline the reviewer's suggestion.

Line 61 to 66

“Prescribing a home-based exercise program without the possibility of check the exercise execution pattern or without a clear element of motivation for compliance can negatively affect clinical results. These factors (the lack of supervision, feedback with the therapist, and lack of adherence to treatment) have been strongly considered possible determinants of poor clinical results for home exercise programs in patients with shoulder pain [9]”.

 [9] Bailey, D.L.; Holden, M.A.; Foster, N.E.; Quicke, J.G.; Haywood, K.L.; Bishop, A. Defining adherence to therapeutic exercise for musculoskeletal pain: a systematic review. Br J Sports Med. 2020, 54,6,326-331. doi: 10.1136/bjsports-2017-098742.

Line 86 to 89

“This platform facilitated the exercise programming with an interesting design for the patient, and also provides feedback between the user and the physiotherapist in real time. These characteristics are differentiating elements from other current rehabilitation methods”.

  1. The statistical analysis section is thorough. Consider adding more details on the clinical significance of the findings, not just the statistical significance.

We have included, in table 3, the effect size values for the analysis using Hedges g approach. In addition, we have added information about their calculation in the statistics section, as shown below, with a new reference.

Line 257 to 259

“Effect size (ES) was estimated using Hedges g [50], with the following scale to categorize the magnitude of this ES: < 0.2 = trivial; 0.2–0.5 = small; 0.5–0.8 = medium; 0.8–1.3 large; and > 1.3 very large [29]”.

[29] Brydges, C.R. Effect Size Guidelines, Sample Size Calculations, and Statistical Power in Gerontology. Innov Aging. 2019,4,3,4. igz036. doi: 10.1093/geroni/igz036.

On the other hand, clinical outcomes have been evaluated using patient-reported outcome measures. The text discusses the clinical relevance of the results obtained for this variable. That is, a minimal clinically important difference for the DASH scale. References published by other authors on the clinical relevance of this result also appear in the text.

Line 463 to 469

“The minimum clinically important difference (MCID) value according to the DASH has been reported as between 10.8 and 15 points [24]. In this study, both the control group and the experimental group achieved a reduction of at least 10 points on the DASH score (10.7 and 11.1 points, respectively), indicating a slight but significant clinical improvement, although not statistically significant. A higher number of patients and a longer follow-up period is required to assess this variation consistently and to check if it is maintained in the long term”.

  1. The description of the eFisioTrack system is comprehensive. Consider adding a figure or diagram to represent the system and its components visually.

In accordance with the reviewer's comment, we have increased the information in Figure 2.

Lines 175-178

“Figure 2. Example of the commonly prescribed and selected exercises. (a) Active bilateral shoulder flexion; (b)(c) Internal and external rotation resisted exercises with elastic bands; (d) Active shoulder abduction; (e) general view of the eFisioTrack scenario; (f) isometric shoulder extension”.

In addition, we have added two news figures (figure 3 and 4) in the section 2.6. eFisioTrack platform, and we have explained it in the text as shown below.

Line 199

“Figure 3 represents visually the system components”-

Line 238 to 240

“A count of repetitions to be performed was shown on a screen, and the system also showed some messages to increase the patient's motivation during the session, in order to achieve maximal adherence and correct execution (Figure 4)”.

  1. The chosen outcome measures are appropriate. It would be beneficial to provide a brief explanation of why these specific measures were chosen.

A brief description of the reasons for the selection of two of the three outcome measures used was included in the original manuscript and we consider that it provides an adequate and brief justification of the clinical relevance of these variables.

Regarding the Disabilities of Arm, Shoulder and Hand (DASH) score and the Constant-Murley (CM) score, as noted in section 2.7. "Outcomes":

lines 275 and 276:"This instrument have been used in previous studies involving physical therapy and exercise for shoulder injuries [24]”

Lines 287-288.  In addition, the use of CM ”… has been specifically validated in shoulder arthroplasty, rotator cuff repair, adhesive capsulitis, and fractures of the proximal humerus, but not in shoulder instability [26]".

However, following the reviewer's suggestion, we have added a brief explanation of why the Short Form-36 Health Survey (SF-36) was chosen, with a new reference.

Line 242 to 245

“The SF-36 is a generic measure of health-related quality of life, with 36 questions. It yields an 8-scale profile of functional health and well-being scores that can be divided in two categories - mental and physical. The value range is between 0 and 100 (higher scores indicating better health status) and has been validated in Spanish [28]”.

  1. Discuss the broader implications of the study for clinical practice and future research. Consider including a paragraph on the potential economic benefits of the eFisioTrack system.

We have added some lines related to this suggestion.

Despite the lack of studies with the Wii remote pad for rehabilitation of shoulder injuries, most of them have been executed on painful hemiplegic shoulder in stroke patients [17,18], we believe that it is an adequate, inexpensive, and easy-to-use device that, associated with the appropriate platform, can offer a good telerehabilitation service. Cost analysis for the eFisioTrack should be considered in the future, as has been done in other pathologies such as low back pain, hip and knee osteoarthritis, and total knee arthroplasty [48-52]….

Line 572 to 583

…In these studies, the use of telerehabilitation shown positive clinical outcomes, with a fewer admission to hospitals, shorter length of stay in hospitals, lower number of visits to the rehabilitation service and, consequently, lower costs.

Given the high prevalence of shoulder injuries, telerehabilitation based on the eFisio track system represents an opportunity to improve clinical and economic data. In addition, the extensive use of the system as a complement to a home therapeutic exercise program could have a positive impact on adherence to performing the exercises. This aspect could potentially improve clinical and functional results.

If this were achieved, its implementation could contribute to reducing costs in the rehabilitation of orthopedic shoulder injuries through: fewer patient trips to the hospital environment; a lower consumption of human resources dedicated to supervising the execution of the exercises”.

“In these studies, the use of telerehabilitation shown positive clinical outcomes, with a fewer admission to hospitals, shorter length of stay in hospitals, lower number of visits to the rehabilitation service and, consequently, lower costs.

Given the high prevalence of shoulder injuries, telerehabilitation based on the eFisio Track system represents an opportunity to improve clinical and economic data. In addition, the extensive use of the system as a complement to a home therapeutic exercise program could have a positive impact on adherence to performing the exercises. This aspect could potentially improve clinical and functional results.

If this were achieved, its implementation could contribute to reducing costs in the rehabilitation of orthopedic shoulder injuries through: fewer patient trips to the hospital environment; a lower consumption of human resources dedicated to supervising the execution of the exercises”.

Extra clarification

After submitting the manuscript to your journal, we received an email from Ms. Elsie Zhao - Managing Editor - advising us to review the current Data Availability Statement in our manuscript and make the necessary adjustments to guarantee the quality and transparency of the research published in Sensors.

In this sense, we have added a file in supplementary material with the database of our study, and we have modified this statement in the manuscript, as shown below.

Line 631 to 632

Data Availability Statement: The original contributions presented in the study are included in the article/supplementary material, further inquiries can be directed to the corresponding author/s.

Round 2

Reviewer 1 Report

Comments and Suggestions for Authors

The author carefully revised and explained the comments of the previous review. The work in this paper is more complete, so I suggest that it be published in Sensors journal. It is expected that the author's subsequent work will continue to supplement and improve this paper.